# A New Generation of Ultrasmall Nanoparticles Inducing Sensitization to Irradiation and Copper Depletion to Overcome Radioresistant and Invasive Cancers

**DOI:** 10.3390/pharmaceutics14040814

**Published:** 2022-04-07

**Authors:** Paul Rocchi, Delphine Brichart-Vernos, François Lux, Isabelle Morfin, Laurent David, Claire Rodriguez-Lafrasse, Olivier Tillement

**Affiliations:** 1Institut Lumière Matière, Université Claude Bernard Lyon 1CNRS UMR 5306, 69622 Villeurbanne, France; rocchi@nhtheraguix.com (P.R.); delphine.brichart@univ-lyon1.fr (D.B.-V.); olivier.tillement@univ-lyon1.fr (O.T.); 2NH TherAguix SA, 38240 Meylan, France; 3Cellular and Molecular Radiobiology Laboratory, Lyon-Sud Medical School, UMR CNRS 5822/IP2I, Univ. Lyon, Lyon 1 University, 69921 Oullins, France; claire.lafrasse-rodriguez@univ-lyon1.fr; 4Institut Universitaire de France (IUF), 75000 Paris, France; 5LiPhy, Université Grenoble Alpes, CNRS, UMR UMR5588, 38401 Grenoble, France; isabelle.morfin@univ-grenoble-alpes.fr; 6Ingénierie des Matériaux Polymères, Université de Lyon, Université Claude Bernard Lyon 1, Université Jean Monet, Institut National des Sciences Appliquées de Lyon, CNRS, UMR 5223, 15, bd A. Latarjet, 69622 Villeurbanne, France; laurent.david@univ-lyon1.fr; 7Department of Biochemistry and Molecular Biology, Lyon-Sud Hospital, Hospices Civils of Lyon, 69310 Pierre-Bénite, France

**Keywords:** copper depletion, ultrasmall nanoparticle, radiosensitization

## Abstract

An emerging target to overcome cancer resistance to treatments is copper, which is upregulated in a wide variety of tumors and may be associated with cancer progression and metastases. The aim of this study was to develop a multimodal ultrasmall nanoparticle, CuPRiX, based on the clinical AGuIX nanoparticle made of the polysiloxane matrix on which gadolinium chelates are grafted. Such hybrid nanoparticles allow: (i) a localized depletion of copper in tumors to prevent tumor cell dissemination and metastasis formation and (ii) an increased sensitivity of the tumor to radiotherapy (RT) due to the presence of high Z gadolinium (Gd) atoms. CuPRiX nanoparticles are obtained by controlled acidification of AGuIX nanoparticles. They were evaluated in vitro on two cancer cell lines (lung and head and neck) using the scratch-wound assay and clonogenic cell survival assay. They were able to reduce cell migration and invasion and displayed radiosensitizing properties.

## 1. Introduction

Copper is a trace metal playing a key role in a wide variety of metabolic processes. An analysis of the human proteome allowed the identification of 54 copper-binding proteins involved in endogenous processes such as mitochondrial respiration, antioxidant defense and extracellular matrix cross-linking [1]. Copper levels must be finely regulated to maintain homeostasis for proper cell function. Any disruption of this balance can lead to sicknesses such as Wilson’s disease (copper overload) and Menkes disease (copper deficiency) or, as described in different preclinical and clinical works, to cancer progression [2,3,4].

Copper may play a role in tumor growth, epithelial-to-mesenchymal transition (EMT), stabilization of the tumor microenvironment [5] and pre-metastasis niche formation [6]. It can also induce a pro-angiogenic response, which is of great importance since tumor size is limited to 1–2 mm in the absence of new blood vessel formation. Copper is also an essential cofactor of various metalloproteins known to be closely correlated with tumor growth and metastatic invasion (Mitogen-Activated Protein Kinase Kinase 1 (MAP2K1), lysyl oxidase (LOX) and lysyl oxidase-like, (LOXL), Secreted Protein Acidic and Cysteine Rich (SPARC), …). Interestingly, higher copper levels were detected in the tumors or serum of patients and animals in many different types of cancers, including oral, breast, prostate, and lung cancers [7]. It is also important to note that circulating copper levels tend to be higher in patients with metastases than in those with localized tumors [8]. This suggests that a disruption of Cu homeostasis may contribute to carcinogenesis as well as to the development of metastases, which thus makes it a prime target for more effective cancer treatment [9].

It is from its involvement in such crucial mechanisms that the idea of removing copper using copper chelators emerged a few decades ago in order to fight cancer. The first major preclinical study was published in 1990 [10], followed by others revealing that a copper depletion strategy could hinder cancer progression. These findings led to clinical trials using chelating agents such as trientine (TETA), tetrathiomolybdate (TM) and D-penicillamine (D-Pen), typically used for the treatment of Wilson’s disease. In all cases, serum copper levels decreased and a phase 2 study on triple negative breast cancer patients at high risk of recurrence (TNBC) treated with TM showed very promising results [11,12]. In addition to successfully depleting copper, they correlated lower levels of copper with reduced levels of endothelial progenitor cells and LOXL-2, both of which are key elements of tumor progression. However, in some trials, using trientine, side effects ranging from anemia to neurological disorders have been observed [13].These disturbances could be related to a non-localized copper depletion.

The use of a copper chelator is now being investigated as part of a multimodal approach, as it might slow cancer progression and enhance tumor cell sensitivity to other treatments [11,14]. Copper chelators, such as TM, are now being combined with chemotherapeutic agents [14,15], immunotherapy treatments [16,17] or radiotherapy [18].

In this approach, the use of nanoparticles could bring many advantages. First, nanoparticles may have a specific biodistribution driven by their physicochemical characteristics [19], which allows copper chelation to be brought closer to the tumor area, where the copper concentration is higher. This approach has been explored by R. Tremmel et al., encapsulating the copper chelator TETA in surface-modified liposome used as nanocarrier [20]. Other teams have developed synthetic nanoparticles enabling copper depletion by themselves through different mechanisms, such as polymer chelation [21,22] and copper–sulfur interaction [23]. Only a few teams took advantage of this specific nanoparticle biodistribution to combine copper depletion and delivery of therapeutic agents at the same time. In this view, P. Zhou et al. developed a pH-sensitive biodegradable polymeric nanoparticle capable of delivering both a copper chelator and the immunotherapeutic agent R848 [24]. In vitro, this nanoparticle showed greater efficiency against metastatic breast cancer than either the copper chelator or R848 alone.

To our knowledge, there is currently no work on nanoparticles combining copper chelation and radiosensitization. In this work, we present a new generation of radio-enhancing drug candidates based on a derivative of AGuIX^®^ nanoparticles currently in phase 2 clinical trials [25,26]. This optimized nanoparticle, named CuPRiX, is expected to have the same initial properties as AGuIX^®^, namely (i) specific accumulation in tumors via the Enhanced Permeability Effect (EPR), (ii) biodegradability and renal excretion, due to its small size, (iii) its characteristics as an MRI contrast agent and (iv) its ability to increase the efficacy of radiation therapy, thanks to the presence of gadolinium [27,28]. To those properties, we aim to add the benefits of copper chelation by generating free chelators (DOTAGA) on the surface of the nanoparticle through the neutralization of their carboxylic moieties and the partial release of gadolinium at a low pH and a relatively high temperature. The efficacy of this newly formed product, called CuPRiX, on cell migration, invasion and radiosensitization has been evaluated in vitro on two cancer cell lines (oral and lung) as a first proof of concept.

## 2. Materials and Methods

### 2.1. Chemical Materials

The starting AGuIX^®^ nanoparticles (gadolinium-chelated polysiloxane nanoparticles) were provided by NH TherAguix (Grenoble, France) as lyophilized powder. These nanoparticles are composed of a polysiloxane matrix on which DOTAGA(Gd) chelates are covalently grafted. Their synthesis has been already described in Le Duc et al. [29]. The 2 M HCl solution was prepared by diluting concentrated HCl (extra pure, 37%) purchased from CarlRoth. The NaOH 1 M solution was prepared using NaOH pellets from Fisher Chemical (Waltham, MA, USA). The Cu^2+^ titration solution was prepared by dissolving CuCl_2_ (powder, 99%) from Sigma Aldrich (Saint-Quentin Fallavier, France) in pH 2 aqueous solution. For the HPLC-UV phase preparation, acetonitrile (CH3CN, ACN, >99.9%) was purchased from Sigma-Aldrich (France) and trifluoroacetic acid (TFA) from Fischer Chemical (Waltham, MA, USA).

### 2.2. CuPRiX_(x)_ Synthesis

AGuIX^®^ nanoparticles were dispersed at 200 g L^−1^ in Milli-Q water (conductivity ρ > 18 MΩ·cm). After 30 min of dispersion, the AGuIX^®^ solution was poured in an HCl 2 M solution. Thus, the final mix composition is 1 M HCl and 100 g L^−1^ in AGuIX^®^. The mixture was left at 50 °C under stirring. A sample of the mixture was extracted every hour in order to follow the gadolinium release by HPLC-IC/MS. After 3 h of reaction, a part of the mix was isolated and the free gadolinium was removed during a purification step performed with the Vivaflow 200 ultrafiltration system (PES membrane, molecular weight threshold = 5 kDa, Sartorius Stedim Biotech, Göttingen, Germany). The final pH was adjusted to 7.4 using 1 M NaOH. The solution was sterile filtered through a 0.2 µm filter to remove the largest impurities. It was then freeze-dried for storage, using Christ-Alpha 1–2 lyophilizer (Coueron, France). This first product will be called CuPRiX_1_. In order to release more gadolinium, the rest of the initial mix was left at 50 °C and under stirring for an additional hour. The same process was followed, which led to a second product called CuPRiX_2_. To summarize the procedure, CuPRiX_1_ and CuPRiX_2_ are obtained by reaction at 100 g L^−1^ in HCl (1 M) during 3 h and 4 h, respectively, before purification by tangential filtration.

### 2.3. Gadolinium Release Followed by HPLC-ICP/MS

In-process extracts were analyzed to follow the gadolinium release in the reaction mixture. The analysis was performed with a Nexion 2000B (Perkin-Elmer, Villebon Sur Yvette, France), coupled with a Flexar LC system (Perkin-Elmer). The separation was performed using a C4 reverse phase column (Jupiter^®^, 5 µm, 300 A, 150 × 4.6 mm, Phenomenex, Le Pecq, France). The measurements were performed on isocratic mode using the following phase composition: H_2_O/ACN/TFA (98.9%/1%/0.1%) at 1 mL/min speed flow. Gd signal was monitored using the isotope 152. The operating conditions used for the ICP-MS were: nebulizer gas flow rate, 0.84 L/min; plasma gas flow rate, 15 L/min; auxiliary gas flow rate, 1.2 L/min; radiofrequency power, 1600 W for the plasma. All other parameters were tuned to maximize the Gd signal. Syngistix software version 2.3 (Perkin Elmer, Villebon Sur Yvette, France) was used to control the ICP-MS. The Gd signal was acquired through the Empower software version 7.3 (Waters, Milford, MA, USA).

### 2.4. Final Measurement of Gadolinium Amount by ICP-MS

The amount of gadolinium in AGuIX^®^_,_ CuPRiX_1_ and CuPRiX_2_ was measured by ICP/MS (Nexion 2000B, Perkin-Elmer, Villebon Sur Yvette, France) with a direct injection mode. The calibration points and the samples were prepared in 1% HNO_3_ solution. Gd signal was monitored following isotopes 158 and 160. The operating conditions used for the ICP-MS were: nebulizer gas flow rate, 0.84 L/min; plasma gas flow rate, 15 L/min; auxiliary gas flow rate, 1.2 L/min; radiofrequency power, 1600 W for the plasma. All other parameters were tuned to maximize the Gd signal. Syngistix software version 2.3 was used to control the ICP-MS. The Gd signal was acquired through Empower software version 7.3.

### 2.5. HPLC-UV

Shimadzu Prominence series UFLC system (Lyon, France), equipped with a CBM-20A controller bus module, a LC-20 AD liquid chromatograph, a CTO-20A column oven and an SPD-20A UV-visible detector was used for Gradient HPLC analysis. Wavelength was fixed at 295 nm for UV-VIS detection. The separation was performed using a C4 reverse-phase column (Jupiter^®^, 5 µm, 300 A, 150 × 4.6 mm) at a flow rate of 1 mL min^−1^. The gradient initial solution is 95% solvent A − 5% solvent B (A = H_2_O/ACN/TFA: 98.9 v%/1 v%/0.1 v%, B = ACN/H_2_O/TFA: 89.9 v%/10 v%/0.1 v%) over 5 min. In a second step, the samples were eluted by a gradient developed from 5 to 90% of solvent B in solvent A over 15 min. The concentration of solvent B was maintained over 5 min. Then, the concentration of solvent B was decreased to 5% over a period of 5 min to re-equilibrate the system, followed by an additional 5 min at this final concentration.

### 2.6. Measurements of Uncomplexed DOTAGA Groups Based on the Formation of DOTAGA@(Cu^2+^)

The amount of uncomplexed DOTAGA groups in CuPRiX_1_ and CuPRiX_2_ was determined by titration by recording the increase in absorbance at 295 nm due to the formation of DOTAGA@(Cu^2+^) complex. A series of samples with a fixed amount of CuPRiX product and an increasing amount of Cu^2+^ was prepared. All the samples were prepared in a pH 4.5 acetate buffer and allowed to react for 30 min to ensure a good complexation. Finally, the samples were injected following the HPLC-UV method previously described. The breaks in slope in the absorbance increase were directly related to the amount of uncomplexed DOTAGA in the product.

### 2.7. Dynamic Light Scattering (DLS) and ζ-Potential Measurements

Direct measurements of the size distribution of the nanoparticles were performed at 10 g/L via Zetasizer NanoS DLS (Dynamic Light Scattering, laser He-Ne 633 nm, Malvern, Palaiseau, France) from Malvern Instrument. The ζ-potential of the nanoparticles was also determined via Zetasizer NanoS. Before the measurement, the nanoparticles were diluted to 10 g/L in an aqueous solution containing 0.01 M NaCl.

### 2.8. Small-Angle X-ray Scattering Measurements

Small-angle synchrotron X-ray scattering (SAXS) was performed at ESRF (Grenoble, France) on a D2AM beamline. The incident photon energy was set to 15.7 keV (i.e., λ = 0.789681 Å^−1^), with a beam size of about 45 µm × 40 µm. CuPRiX_2_ and AGuIX^®^ suspensions were prepared at 100 g/L and transferred in glass tubes (Deutero GmbH, Kastellaun, Germany, ref. 600020-200, external diameter: 3 mm, length: 60 mm, width: 0.2 mm). The sample-to-detector distance was set close to 1.84 m. The D5 solid-state detector (IMXPAD) was used for the 2D image collection. All scattered images were normalized by the transmitted intensity, and radial averages were calculated around the image center (mean center of incident beam). The 1D radially averaged pattern of the empty cell (solvent + tube) was subtracted to the 1D scattered patterns of the samples in order to deduce the net intensity scattered by nanoparticles, i.e., I vs. q = 4π.sin(θ)/λ where 2θ is the scattering angle and q is the scattering vector.

### 2.9. Relaxivity Measurements

Relaxivity measurements were performed at 100 g/L on a Bruker Minispec mq60 NMR analyzer (Bruker, Billerica, MA, USA) at 37 °C at 1.4 T (60 MHz).

### 2.10. Cell Lines and Cell Culture

The human lung adenocarcinoma cell line, A549, was purchased from the European Collection of Authenticated Cell Culture (ECACC 86012804). A549 cells were cultured in F12K medium (Gibco^TM^) supplemented with 10% Fetal Bovine Serum (FBS) (Dominique Dutscher SAS, Bernolsheim, France) and 1% penicillin-streptomycin (Gibco^TM^). The Head and Neck Squamous Cell Carcinoma (HNSCC), SQ20B was established from HNSCC (larynx) tumors and provided by J.B. Little (Department of Cancer Biology, Harvard School of Public Health, Boston, MA, USA). Its sub-population of cancer stem-cells-like (SQ20B-CSCs) was obtained by flow-cytometry cell sorting and cultured as described in [30,31]. Briefly, cells were first sorted by the Hoechst Dye Efflux Assay, which led to the generation of a Side Population (SP). Then, from this SP, SQ20B-CSCs were isolated based on their CD44 expression. SQ20B-CSCs were cultured in a mixture of DMEM, Ham’s F12 (3:1, Gibco^TM^) supplemented with 0.4 µg/mL hydrocortisone (Sigma-Aldrich, Saint-Quentin Fallavier, France), 1% penicillin-streptomycin, 5% FBS and 20 ng/mL Epithelium Growth Factor (EGF, Sigma-Aldrich). Due to their cancer stem-cell-like characteristics, only the SQ20B-CSC subpopulation was used. Both cell lines were cultured in humidified atmosphere with 5% CO_2_, at 37 °C.

### 2.11. Scratch-Wound Assay

Cells (4 × 10^4^/well) were seeded in IncuCyte^®^ ImageLock 96-well plates (Essen BioScience, Ltd., Royston, UK) and incubated for 16 h at 37 °C to reach 90–100% confluency. The cell layer was then scratched with the 96-well Wound-Maker^TM^ (Essen BioScience) and washed 2 times with PBS 1X (Gibco). For invasion assays, 50 µL of reduced Matrigel (Corning) diluted at 1 mg/mL in medium was added to each well and the plate was incubated 30 min at 37 °C to allow the gelation of the matrix. Afterwards, 100 µL of medium alone or containing CuPRiX_2_ (800 µM of DOTAGA(Gd) and 500 µM of uncomplexed DOTAGA) or AGuIX^®^ (800 µM of DOTAGA(Gd)) was added to the appropriate wells. Finally, the plate was placed in the IncuCyte (objective 10X), and images of each well were taken automatically every 2 h in the CO_2_ incubator. Data were analyzed using IncuCyte ZOOM software (v. 2018A, Essen BioScience, Ltd., Royston, UK) and expressed as a percentage of wound confluency (relative wound density).

### 2.12. LOX Activity Assay

A549 and SQ20B-CSC cells were seeded in 25 cm^2^ flasks in appropriate medium and allowed to adhere. The medium was then replaced with medium containing CuPRiX_2_ (800 µM of DOTAGA(Gd) and 500 µM of uncomplexed DOTAGA) or AGuIX^®^ (800 µM of DOTAGA(Gd)) and incubated for 24 h or 48 h. At both time points, the medium from each condition was collected and LOX activity was determined using LOX Activity Fluorometric Assay Kit (ab112139, Abcam, Cambridge, UK) according to the manufacturer’s instructions. Briefly, the cell culture medium was collected and centrifuged at 13,000× *g* for 5 min at 4 °C. Then, 50 µL of supernatant were distributed in duplicate into a clear-bottom 96-well plate (50 µL of assay buffer was used as a blank) and 50 µL of LOX reaction mix was added to each well. The plate was incubated for 40 min at 37 °C in the dark. Fluorescence was then monitored on a microplate reader at Ex/Em = 560/590 nm. LOX activity levels are expressed as relative to background noise (blank control).

### 2.13. Clonogenic Survival Assay

Cells were seeded at a density of 4 × 10^4^ cells/cm^2^ and allowed to grow overnight. The cells were then incubated for 24 h in serum-free media alone or supplemented with AGuIX^®^ (800 µM of DOTAGA(Gd)) or CuPRiX_2_ (800 µM of DOTAGA(Gd) and 500 µM of uncomplexed DOTAGA). Cells were irradiated with single dose of 2, 3, 4 or 6 Gy with 220 kV X-ray at a dose rate of 2 Gy.min^−1^ and then incubated for 4 h. Afterwards, they were washed with PBS, trypsinized, counted, replated in 25 cm^2^ flasks, and incubated for 10 days. Cells were then fixed with ethanol 96% (VWR) and stained with Giemsa (Sigma-Aldrich) diluted at 1/20 in distilled water. The flasks were rinsed with distilled water and allowed to dry overnight. Colonies with 64 cells or more were counted using the Colcount^TM^ system (Oxford Optronix Ltd., Abingdon, United Kingdom). Clonogenic survival curves were fitted according to the linear quadratic equation $\mathrm{SF}={\;e}^{-\left(\alpha D+{\beta D}^{2}\right)}$ where SF is the surviving fraction; α  represents the probability of lethal event, β the sublethal events and D the irradiation dose.

## 3. Results and Discussion

### 3.1. From AGuIX^®^ to CuPRiX: Chemically Designed to Increase the Amount of Uncomplexed DOTAGA

#### 3.1.1. Controlled Release of Gadolinium

AGuIX^®^ nanoparticles are ultrasmall nanoparticles made of polysiloxane core and covalently grafted gadolinium chelates. The chelate is DOTAGA (2-(4,7,10-tris(carboxymethyl)-1,4,7,10-tetraazacyclododecan-1-yl)pentanedioic acid), a derivative from DOTA (1,4,7,10-tetraazacyclododecane-1,4,7,10-tetraacetic acid). DOTA is a macrocyclic molecule well known for the high stability constants of its complexes formed with trivalent lanthanides cations such as Gd^3+^ with log_10_(K) = 25.6 [32]. DOTAGA has been chosen due to its additional carboxylic function that ensure strong gadolinium chelation after functionalization on the polysiloxane inorganic matrix (log K = 24.78) [33]. This high stability is mainly due to the carboxylic acid groups capable of chelating in the deprotonated state [34]. Here, the dissociation of gadolinium from the DOTAGA@(Gd^3+^) groups covalently grafted on AGuIX^®^ has been performed through protonation of the chelate in strong acidic conditions. Indeed, AGuIX^®^ nanoparticles were placed in a 1 M HCl acidic medium in order to force the competitive protonation of the chelate, leading to the release of some of the Gd^3+^ cations [35,36]. The dissociation kinetics was favored by heating the solution at 50 °C during the release step. The free Gd^3+^ ions can be easily separated from the rest of the mixture components by HPLC using a reverse phase C4 column. Thanks to the quantification by ICP-MS, the released gadolinium could be monitored all along the process, as illustrated in Figure 1. In the end, the amount of uncomplexed DOTAGA in the final product should be directly related to the amount of gadolinium initially released at the time we decide to stop the reaction. Two different nanoparticles, CuPRIX_1_ and CUPRiX_2_, with an increased ratio of uncomplexed DOTAGA have been prepared by ending the reaction after 3 and 4 h, respectively. To prevent the released Gd^3+^ from being chelated again and to ensure a good safety of the product regarding the toxicity of free Gd^3+^ [37], a purification step has been performed using Vivaflow 200 ultrafiltration system with a cutoff of 5 kDa. The absence of free Gd^3+^ in the mixture after purification and before freeze-drying has been confirmed by HPLC-ICP/MS, as shown on Appendix A.

#### 3.1.2. Characterization of CuPRiX_1_ and CuPRiX_2_

##### Size

The size of the nanoparticles will impact their pharmacokinetics [19] as well as its elimination pathway [38]. The hydrodynamic diameter (D_H_) of the particles has been measured through dynamic light scattering (DLS). The volume distributions of the hydrodynamic diameter for CuPRiX_1_ and CuPRiX_2_ are displayed in Figure 2c. As expected, the measured D_H_ are close, equal to 4.6 ± 1.6 nm for CuPRiX_1_ and 5 ± 2.1 nm for CuPRiX_2_, respectively. These diameters are below the glomerular cutoff (<8 nm) [38] and thus good renal clearance from the body can be expected, as previously observed for AGuIX^®^ nanoparticles [39]. Moreover, they are also close to the diameter of AGuIX^®^ nanoparticles (D_H_ = 3.6 ± 1.3 nm, displayed on Appendix A). The gyration radii of CuPRIX2 and AGuIX^®^ nanoparticles was also evaluated by SAXS. The Guinier plots are shown in Appendix A and yield gyration radii of 17.9 and 19.0 Å for CuPRIX_2_ and AGuIX^®^, respectively. In log–log representation (see Appendix A), the theoretical Guinier law (i.e., *I*(*q*) = *I*_o_·exp(−*q*^2^·*R*_g_^2^/3)) deviates from the experimental results at high *q* values, as expected from the transition to surface scattering with a modified Porod’s law [40] (*I*(*q*) = *C/q*^4^ − *B/q*^2^), accounting for diffuse boundary. Indeed, the slope of the scattering diagrams in the log–log scale is slightly higher than 4. In the low-*q* range, scattering by AGuIX^®^ displays a scattering excess compared to the theoretical Guinier law, showing the presence of small aggregates of particles. An opposite trend is observed for CuPRIX_2_. This may originate from interparticle repulsive interactions associated with a negative charge surface, resulting from the partial dissociation of DOTAGA-Gd^3+^ complexes and thus the regeneration of the negatively charged uncomplexed DOTAGA. 

##### HPLC-UV

The HPLC-UV chromatograms recorded at 295 nm, which correspond to the maximum absorption of the initial nanoparticle AGuIX^®^ [41], can be found in Figure 2b. High purity close to 99% is obtained for CuPRiX_1_ and CuPRiX_2_ products measured directly from the absorbance ratio (purity% = Abs[10–15 min]/Abs[0–15 min]) of the HPLC-UV chromatograms. In addition, both modified products exhibit an elution peak located at a similar retention time (Tr) of 10.9 min, which is also close to that of AGuIX^®^ equal to 10.7 min (Appendix A).

##### ζ-Potential Measurement

The recorded ζ-potential of CuPRiX_1_ and CuPRiX_2_ depending on the pH is presented in Figure 2d. The nanoparticles share the same general behavior towards pH. The particles are positively charged at low pH due the protonated amine groups on the surface coming from the initial APTES used in the synthesis of AGuIX^®^ [33]. At pH 3, the predominant form of the uncomplexed DOTAGA groups is the neutral one, H_4_DOTA [42]. With increasing pH, the amount of the NH_3_^+^ decreases and the DOTAGA groups increasingly switch to the negative charged form, leading to a decrease in ζ-potential. The pH at the Iso-Electric Point (IEP) decreases from 7.2 for AGuIX^®^ (Appendix A) to 6.7 for CuPRiX_1_ and 6.3 for CuPRiX_2_. This decrease in IEP is in agreement with the increase in the amount of uncomplexed DOTAGA groups. Indeed, complexed DOTAGA@(Gd^3+^) is more stable and has a fixed negative charge of −1 compared to uncomplexed DOTAGA which tends to −4 at high pH. For an equivalent amount of amine groups, the IEP will therefore be obtained at a lower pH.

##### Uncomplexed DOTAGA Measurement

The amount of uncomplexed DOTAGA groups in CuPRiX_1_ and CuPRiX_2_ was measured by HPLC-UV. Different samples have been prepared with a fixed number of particles and an increasing amount of Cu^2+^ at fixed pH to ensure increasing complexation. Due to the high absorbance of the DOTAGA@(Cu^2+^) complex at 295 nm [36], a significant increase in the overall absorbance can be observed at this wavelength (CuPRiX_1_ measurement is shown in Appendix A, CuPRiX_2_ measurement is shown in Appendix A) until there was no free uncomplexed DOTAGA group ready to chelate. The amounts of uncomplexed DOTAGA groups in CuPRiX_1_ and CuPriX_2_ are approximately 182 nmol and 253 nmol per mg of product, respectively. In comparison, AGuIX^®^ displays only 8 nmol of uncomplexed DOTAGA per mg of product (Appendix A).

##### Relation between Gadolinium Content and Relaxivity Measurements

Finally, the amount of remaining gadolinium was measured by ICP/MS. The results can be found in Table 1 and show a decrease in gadolinium content with time spent in the acidic medium (10.5%, 8.2% and 6.2% for AGuIX^®^, CuPRiX_1_ and CuPRiX_2_, respectively). The total amount of DOTAGA in each product (DOTAGA(Gd) + Uncomplexed DOTAGA) can be then calculated.
[total DOTAGA] = [Uncomplexed DOTAGA] + [Gd]
Uncomplexed DOTAGA content (%) = [Uncomplexed DOTAGA]/[total DOTAGA]

AGuIX^®^ has a total DOTAGA content of 676 nmol per mg (1.2% of Uncomplexed DOTAGA), CuPRiX_1_ has a total DOTAGA content of 703 nmol per mg (25.9% of Uuncomplexed DOTAGA) and CuPRiX_2_ has a total DOTAGA content of 647 nmol per mg of product (39.1% of uncomplexed DOTAGA). As a conclusion, with a variation of about ±4% from the initial total DOTAGA content of AGuIX^®^, CuPRiX_1_ and CuPRiX_2_ show a similar total DOTAGA content to that of the starting nanoparticle (See Table 1) and the acidic treatment does not induce strong hydrolysis of the inorganic matrix. CuPRiX_1_ and CuPriX_2_ show higher r_1_ and r_2_ than AGuIX^®^, 0which may be explained by higher hydrodynamic diameter correlated to higher correlation rotation time [27], even if this hypothesis is mitigated by a comparable radius of gyration determined by SAXS for AGuIX^®^ and CuPRiX_2_ (See Appendix A).

### 3.2. CuPRiX Impacts Migration Processes

With copper being a key element in metastasis formation, the effects of its chelation on cell migration were evaluated using a wound-healing assay.

#### 3.2.1. Determination of the Optimal Dose for the Use of CuPRiX

In order to determine the optimal dose of both CuPRiX_1_ and CuPRiX_2_ to be used in the following assays, a first evaluation of cell migration was performed with concentrations ranging from 100 to 500 µM of non-chelated DOTAGA. Maximum efficacy was observed for a concentration of 500 µM (See Appendix A). It can also be noted that for an equal concentration of uncomplexed DOTAGA, CuPRiX_1_ and CuPRiX_2_ produce the same effect on cell migration (See Appendix A). This indicates that both types of CuPRiX can be used indiscriminately. A concentration of 500 µM of CuPRiX_2_ was selected for all further assays in order to (i) maximize efficiency and (ii) limit the concentration of nanoparticles as much as possible to avoid possible side effects linked to the concentration of nanoparticles. 

#### 3.2.2. CuPRiX Decreases Migration and Invasion of A549 and SQ20B-CSCs Cells

Figure 3 shows representative images of A549 (See Figure 3a) and SQ20B-CSC (see Figure 3b) cell migration in the absence or in presence of CuPRiX_2_ as well as the relative wound density calculated after 12, 24, 36 and 48 h of treatment, with (see Figure 3e,f) or without Matrigel (See Figure 3c,d). CuPRiX_2_ was able to significantly decrease the motility of A549 and SQ20B-CSCs cells. At 48 h after wounding without Matrigel, the wound was closed at 80.6 ± 8.6% and 96.7 ± 3.4% for A549 and SQ20B-CSCs cells, respectively, whereas only 68.7 ± 6.6% and 85.8 ± 8.4% of the wound was closed for treated cells (*** *p* < 0.001, see Figure 3c). When A549 and SQ20B-CSCs cells were treated with CuPRiX_2_, in the presence of Matrigel, the invasion process was decreased, as 56.8 ± 5.8% and 72.2 ± 7.4% of the wound was closed, respectively. These values are lower than for untreated cells where 76.4 ± 8% and 86.1 ± 7.9% of the wound was closed after 48 h (** *p* < 0.01 for A549,*** *p* < 0.001 for SQ20B-CSC, See Figure 3e,f). For both cell lines, AGuIX^®^, at the same concentration of gadolinium, did not interfere with migration or invasion, demonstrating the distinctive efficacy of CuPRiX and uncomplexed DOTAGA.

The difference in wound closure cannot be explained by cell growth arrest because CuPRiX did not affect the proliferation of A549 and SQ20B-CSCs cells (see Appendix A), and this shows that CuPRiX_2_ can decrease the migration and invasion of A549 and SQ20B-CSC cells.

#### 3.2.3. CuPRiX Decreases LOX Activity

After showing the ability of CuPRiX to decrease cell migration and invasion with a functional assay, the objective of this experiment was to ensure that this effect could be attributed to copper chelation. To this end, the activity of lysyl oxidase (LOX), a copper-dependent enzyme involved in migration pathways, was evaluated after the addition of CuPRiX_2_ or AGuIX^®^. Treatment for 24 h and 48 h with CuPRiX_2_ resulted in a 26.4% (** *p* < 0.01) and 33.2% (*** *p* < 0.001) decrease in LOX activity for A549 cells (Figure 4a) and 21.7% (*** *p* < 0.001) and 23.4% (*** *p* < 0.001) for SQ20B-CSC (Figure 4b) cells, respectively. AGuIX^®^ did not alter LOX activity in either cell line. 

The decrease in LOX activity could be the consequence of reduced copper availability, showing that CuPRiX_2_ was able to chelate copper and make it unavailable to LOX. In addition, the ability of CuPRiX_2_ to chelate copper in Cu-laden water has already been proven (see Appendix A), but this assay demonstrates this feature in a biological medium.

### 3.3. CuPRiX Radiosensitizes A549 and SQ20B-CSC

Clonogenic assays were performed to compare the radiosensitizing efficacy of AGuIX^®^ and CuPRiX_2_, both presenting DOTAGA(Gd) on their surface (Figure 5). Radiobiological parameters were determined from these data (Table 2). For both cell lines, cell survival fractions (SF) decreased with CuPRiX_2_ and AGuIX^®^. At 4 Gy, the SF of SQ20B-CSC and A549 alone was 0.33 and 0.15, respectively. When treated with AGuIX^®^ and CuPRiX_2_, the SF of SQ20B-CSC at 4 Gy decreases to 0.26 and 0.24, respectively, and the SF of A549 decreases to 0.12 in both cases. Additionally, the probability of having a direct lethal effect, represented with the α coefficient, increased with both CuPRiX and AGuIX^®^. In addition, treatment with CuPRiX_2_ and AGuIX^®^ resulted in a 13% and 18% increase in A549 cell death and an 11% and 9% increase in SQ20B-CSC cell death at 2 Gy, respectively (Table 2). With an equivalent amount of Gd (800 µM), CuPRiX_2_ induces about the same radiosensitizing effect as AGuIX^®^ on both cell lines. 

Coefficients α and β are determined using the linear-quadratic model, with α being the probability of a lethal event occurring and β being the probability of a sublethal event occurring; D_10_ is the calculated dose leading to 10% cell survival fraction; and SER_2Gy_ is the Sensitizing Enhancement Ratio calculated as follow:
$$\mathrm{SER}2\mathrm{Gy}\left(\%\right)=\frac{\mathrm{SF}2\mathrm{Gy}\;\left(\mathrm{control}\right)-\mathrm{SF}2\mathrm{Gy}\;\left({\mathrm{AGuIX}}^{®}\right)}{\mathrm{SF}2\mathrm{Gy}\;\left(\mathrm{control}\right)},$$
with SF_2Gy_ being the survival fraction at 2 Gy.

### 3.4. Discussion

Because elevated levels of copper have been observed in multiple cancer types and are associated with mechanisms such as cancer migration and the development of metastases, there is increasing interest in copper chelation as a cancer therapy [9,11]. In this view, many of the copper chelators tested are oral drugs used for disease such as Wilson’s disease that have been repurposed for cancer treatment. Although they all have their own ability to chelate copper, some copper chelators may have important side effects [9]. An advantage of using nanoparticles as multifunctional macro-chelatants is their ability to target tumors via the EPR effect [43] and potentially induce a blockage of migration mechanisms within the tumor. In this study, the efficacy of CuPRiX nanoparticles to chelate copper was evaluated and was shown to decrease migration on one hand, and to enhance radiation therapy efficacy on the other hand. As increased copper levels have been measured in oral [44] and lung [45,46] cancers and they are both prone to metastasis, we tested one cell line on each cancer: A549 (lung) and SQ20B (oral). We chose to test our drug on a subpopulation of the SQ20B cell line that exhibits stem-cell characteristics such as high radioresistance [47] and migratory and invasive capabilities [48]. 

CuPRiX, unlike AGuIX^®^, was able to decrease cell migration and invasion in our two models, indicating that the presence of free chelate at its surface triggers an effect. In the literature, few articles demonstrate effective copper chelation in vitro, as their ability to chelate copper has been widely tested and validated. As this work is a first proof-of-concept study, and despite the known ability of DOTAGA to chelate copper, it was important to correlate the observed migration inhibition with copper chelation. LOX activity, a copper-dependent enzyme, was significantly decreased in both cell lines when treated with CuPRiX. This confirms the effectiveness of CuPRiX in chelating copper and supports the hypothesis that the effect on migration is indeed due to a lack of copper availability even if the chelation of other metals may also play a role. Additionally, LOX is known to be involved in metastasis formation [49], and it has been shown that catalytically active LOX regulates in vitro motility and cell–matrix adhesion formation [50] and is able to promote tumorigenesis and metastasis [51]. Therefore, CuPRiX, by copper chelation, has shown its potential to inhibit this key enzyme for metastasis and might be able to have a significant impact on cancer progression. To our knowledge, only one nanoparticle was shown to induce a decrease in cell migration by copper chelation, in a human umbilical vein endothelial cell line (HUVEC) [24]. 

The complexity of cancer mechanisms and the heterogeneity of response to treatments make combination therapies a necessary tool to overcome resistance. Few studies have evaluated the effect of copper chelation associated with radiation therapy. In 2006, Khan et al. showed that the combination of TM and radiotherapy improved local control of HNSCC in an isogenic mouse model [18]. Despite these promising results, there are currently no nanoparticles combining copper chelation and radiosensitization in their effect.

CuPRiX is a nanoparticle derived from AGuIX^®^, a radiosensitizing agent currently undergoing evaluation in multiple phase II clinical trials in different types of cancer. The efficacy of AGuIX^®^ as a radiosensitizing agent has been well documented [27,52], and our results showed that CuPRiX_2_ is also able to induce radiosensitization due to the presence of gadolinium. Since the observed sensitizing efficacy is equivalent whether using AGuIX^®^ or CuPRiX with an equivalent amount of gadolinium, this suggests that it is solely due to the presence of gadolinium. Nevertheless, CuPRiX appears to be a strong candidate as an anti-migratory agent. Recently, Yan, We, Hu et al. identified a novel role of copper in promoting the radioresistance of hepatocellular carcinoma cells [53]. We can hypothesize that a synergistic effect of metallic nanoparticles and copper chelation could be obtained in cell lines or tumors with a significantly elevated copper amount. Since CuPRiX is derived from AGuIX^®^ and that display very close physico-chemical features, we can expect similar behaviors in terms of biodistribution, degradability and contrast agent properties [54] in future preclinical and clinical studies.

## 4. Conclusions

In this manuscript, we present a new generation of hybrid nanoparticles, derived from AGuIX^®^, capable of overcoming the radioresistance of tumors, with the presence of DOTAGA(Gd), and potentially limiting metastases with the chelation of copper by uncomplexed DOTAGA that decreases cell migration and invasion in two different cell lines. The chemical synthesis offered the possibility to finely design the amount of non-chelated DOTAGA on the final nanoparticle by controlling reaction time. These nanoparticles display ultrasmall sizes and zeta potentials close to those of AGuIX^®^ nanoparticles. 

As a first proof of concept, we showed that, thanks to the presence of free DOTAGA on its surface, CuPRiX_2_ can successfully chelate copper in a complex medium and decrease the migration and invasion of two lung and oral cancer cell lines. This observation could be partly explained by the inhibition of a copper-dependent enzyme, LOX, involved in metastasis formation. In addition, we confirmed the radiosensitizing efficacy of CuPRiX, in the same way as AGuIX^®^ nanoparticles.

Further experiments are now needed to better understand the underlying mechanisms as well as the efficacy of CuPRiX_2_ on other cancer models in vitro and in vivo.

## 5. Patents

Two patents have been filed on the results described in this publication by PR, FL and OT for the chemical process (FR2011904) and by PR, FL, OT, DBV and CRL for copper chelation in oncology (FR2011903).

## Figures and Tables

**Figure 1 pharmaceutics-14-00814-f001:**
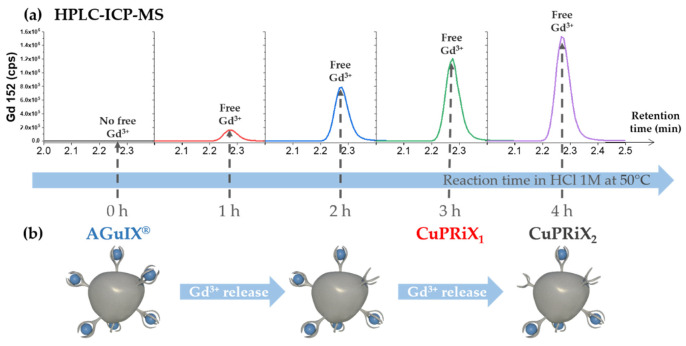
(**a**) Increase in free gadolinium (retention time close to 2.3 min) at different time points of the reaction followed by HPLC—ICP/MS. (**b**) Schematic view of the impact of the process on the nanoparticle starting from AGuIX^®^ to obtain CuPRiX_1_ (3 h of reaction) and CuPRiX_2_ (4 h of reaction).

**Figure 2 pharmaceutics-14-00814-f002:**
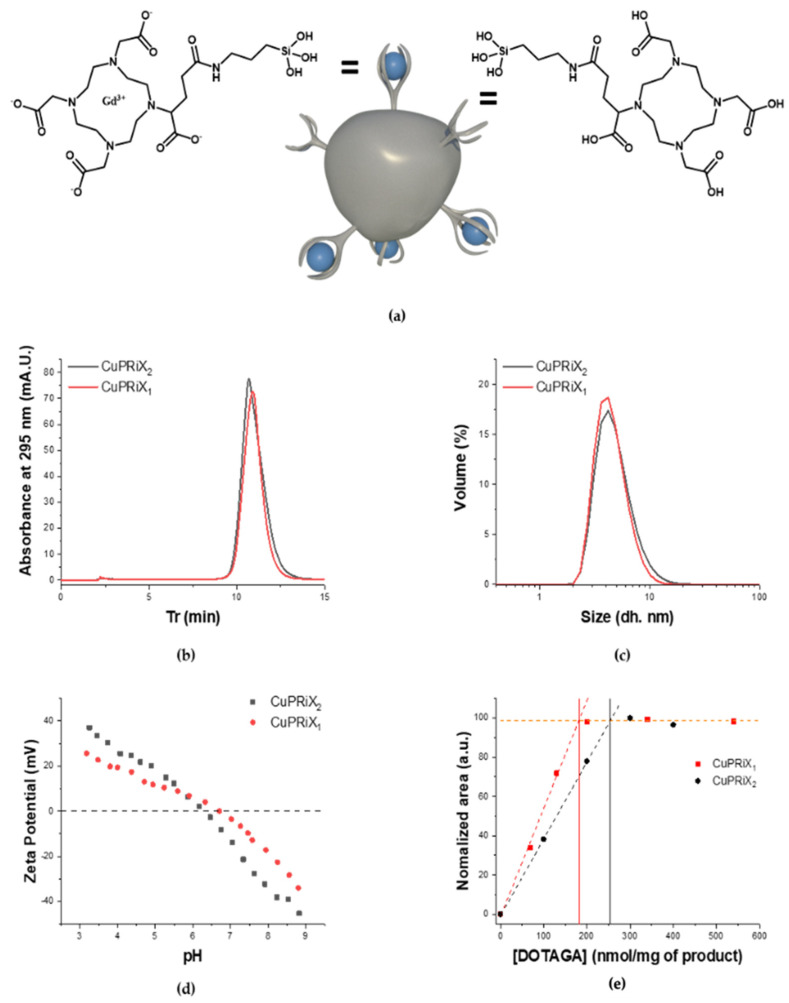
(**a**) Schematic representation of CuPRiX_X_ with a detailed structure of DOTAGA(Gd) and uncomplexed DOTAGA groups; (**b**) HPLC-UV chromatograms of CuPRiX_1_ and CuPRiX_2_ (10 µL, 100 g/L) recorded at 295 nm; (**c**) Hydrodynamic diameter distribution in volume obtained by dynamic light scattering; (**d**) Zeta potential vs. pH for CuPRiX_1_ and CuPRiX_2_; (**e**) Comparison of the amount of DOTA groups between CuPRiX_1_ and CuPRiX_2_ based on their unchelated DOTAGA measurements.

**Figure 3 pharmaceutics-14-00814-f003:**
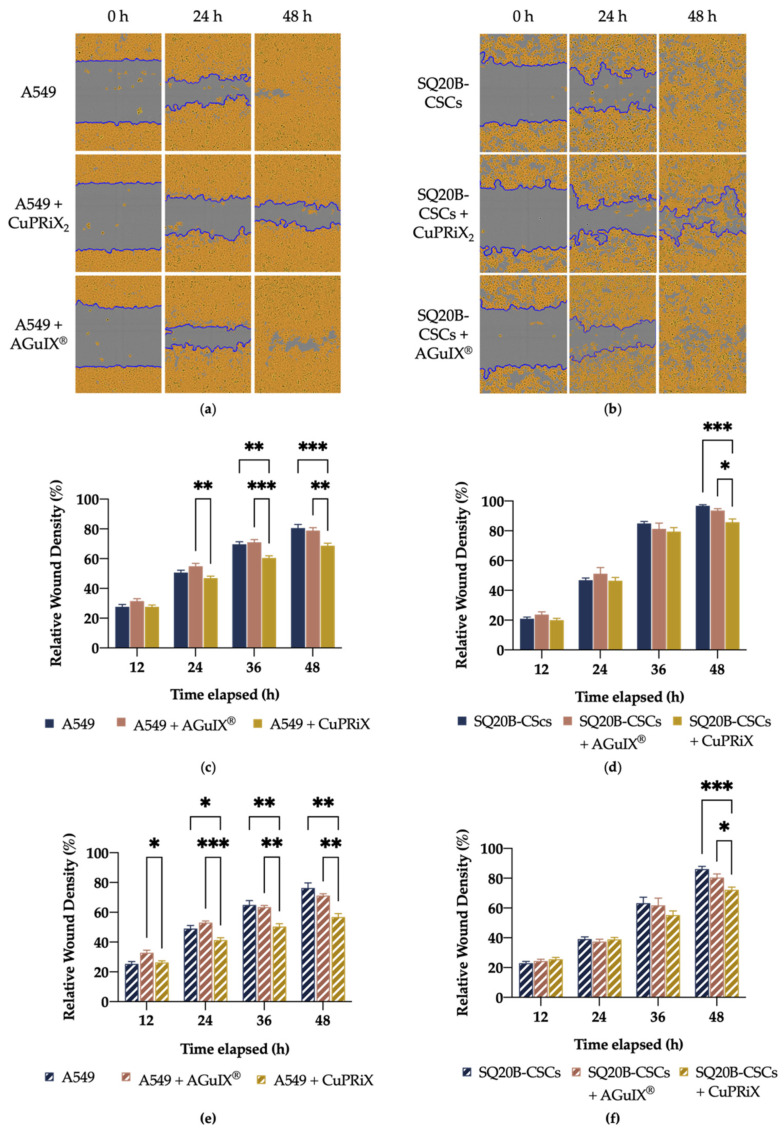
Effect of AGuIX^®^ and CuPRiX_2_ on migration and invasion of A549 and SQ20B-CSCs with the scratch wound assay. Representatives images and quantitative analysis of migration of A549 (**a**), (**c**) and SQ20B-CSCs (**b**,**d**) cells after treatment with CuPRiX_2_ (800 µM of DOTAGA(Gd) and 500 µM of uncomplexed DOTAGA) or AGuIX^®^ (800 µM of DOTAGA(Gd)). (**e**,**f**) Quantitative analysis of invasion (1 mg/mL of Matrigel) of A549 and SQ20B-CSCs cells, respectively, after treatment with CuPRiX_2_ or AGuIX^®^. Relative wound density is a measure of the density of the cell region (%). Data are presented as mean ± SEM (*n* = 3) with 6 technical replicates for each experiment. * *p* < 0.05, ** *p* < 0.01, *** *p* < 0.001, two-way ANOVA.

**Figure 4 pharmaceutics-14-00814-f004:**
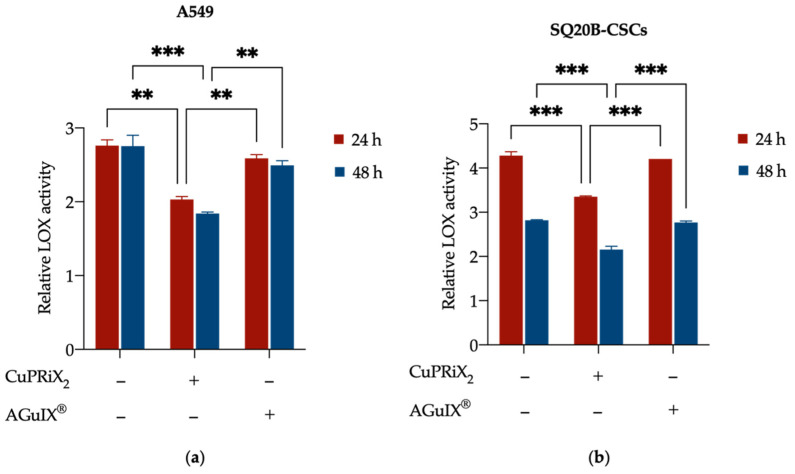
Effect of AGuIX^®^ (800 µM of DOTAGA(Gd)) and CuPRiX_2_ (800 µM of DOTAGA(Gd) and 500 µM of uncomplexed DOTAGA) on LOX activity in (**a**) A549 and (**b**) SQ20B-CSCs cells. Cells were treated for 24 h and 48 h and the activity of LOX (an extracellular copper-dependent enzyme) in the culture medium was assessed. LOX activity levels are relative to background noise (blank control). Data are presented as mean ± SD (*n* = 2) with 2 technical replicates for each experiment. ** *p* < 0.01, *** *p* < 0.001, two-way ANOVA.

**Figure 5 pharmaceutics-14-00814-f005:**
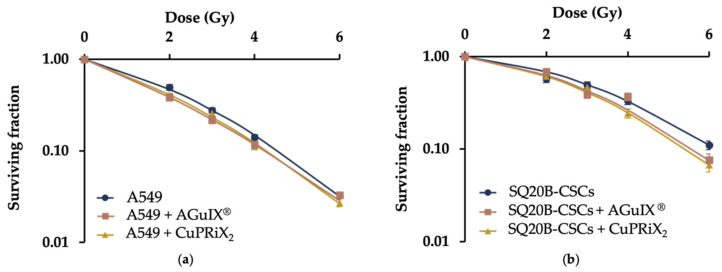
Radiosensitizing effect of AGuIX^®^ (800 µM of DOTAGA(Gd)) and CuPRiX_2_ (800 µM of DOTAGA(Gd) and 500 µM of uncomplexed DOTAGA) on (**a**) A549 and (**b**) SQ20B-CSCs cell lines. Both cell lines were treated with 800 µM of Gd. Each curve is the mean of 3 independent experiments.

**Table 1 pharmaceutics-14-00814-t001:** Comparison of the chemical and physical characteristics of AGuIX^®^, CuPRiX_1_ and CuPRiX_2_.

Product	Gd (w%)	Free DOTAGA (nmol/mg)	Tr (min)	D_H_ (nm)	pH IEP	r_1_ (s^−1^·mM^−1^)	r_2_ (s^−1^·mM^−1^)
AGuIX^®^	10.5	8 ± 2	10.9	3.6 ± 1.34	7.15	18.9	30.4
CuPRiX_1_	8.2	182 ± 14	10.9	4.6 ± 1.56	6.71	20.2	33.5
CuPRiX_2_	6.2	253 ± 8	10.7	5 ± 2.1	6.29	30.7	51.2

**Table 2 pharmaceutics-14-00814-t002:** Summary of radiosensitizing parameters, D_10_ and SER_2Gy_.

Parameters	A549	A549 + CuPRiX_2_	A549 + AGuIX^®^	SQ20B-CSCs	SQ20B-CSCs + CuPRiX_2_	SQ20B-CSCs + AGuIX^®^
α	0.2825	0.3691	0.422	0.1005	0.1477	0.1136
β	0.0485	0.0393	0.0282	0.0444	0.0504	0.0492
D_10_	4.57	4.28	4.25	6.16	5.45	5.6
SER2Gy	-	13%	18%	-	11%	9%

## Data Availability

Not applicable.

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
