# Peer review of "A New Generation of Ultrasmall Nanoparticles Inducing Sensitization to Irradiation and Copper Depletion to Overcome Radioresistant and Invasive Cancers"

_pharmaceutics, 2022, doi:10.3390/pharmaceutics14040814_

Round 1
Reviewer 1 Report
Authors have generated an ultra-small radiosensitizing nanoparticles capable of Cu depletion to overcome radioresistant and invasive tumor. The hypothesis was clear and it represent a relevance of research work carried out. However, there is a need for more robust data by doing some other relevant experiments and by more clearly presenting the existing data before considering it for publication. There’s need to be major revision before considering the manuscript for the publication.
There is some typo- and grammatically mistakes in the text which should be corrected.
Major comments:
- The hypothesis is very clear, approach is fine, however in the synthesis part of CuPRiX nanoparticles, it is not clear how it was obtained or derived from AGuIX nanoparticles. There is a clear need to demonstrate how the synthesis was undertaken, in more clear and stepwise manner in order for it to be reproducible for scientific community. Please rewrite the synthesis part from line 112-124 and expand upon how did author obtain the CuPRiX nanoparticle. Further, just by reading the synthesis it is not immediately obvious how should one obtain CuPRiX nanoparticle using AGuIX, so authors need to focus on clearly demonstrating the synthesis.
- In the name CuPRiX what does PR stands for, also, CuPRiX1 and CuPRiX2 need to be clearly mentioned, abbreviation need to be clear.
- Suddenly authors talk about the DOTAGA groups in the nanoparticles whereas in the synthesis it is not mentioned what was the chemistry behind and how they end up having the group attached or tailored/tagged to the nanoparticles as part of it as a whole. In Method section 2.6. the DOTAGA…line 159 subheading suddenly appears its really confusing, please explain clearly and even better if authors put some perspective about it in their synthesis part itself. There needs to be a separate sub heading with synthesis of DOTAGA tagged nanoparticle synthesis or add separate para in nanoparticle synthesis section.
- Given the ultrasmall size nanoparticle which can be cleared by RES system, how the authors are talking about EPR effect in line 92, please cite reference for the same or correct it.
- Reason for choosing lung cancer and head neck cancer cell with head neck cancer stem cells? What was the most effective dose of radiation +Nanoparticle combination (for both cell lines) In figure 5 was the clonogenic assay done by using single fraction of each radiation dose?
- There is need for more robust data for supporting radiosensitization potential of a synthesized nanoparticle, did author undertake the effect of nanoparticles in ROS generation mediated oxidative stress? It would be interesting to see how nanoparticle improve the ROS formation within the tumor cells.
- Double stranded DNA damage is another important hallmark of radiation mediated tumor cell damage, authors should carry out yH2AX based immunofluorescence assay to study the effect of nanoparticle together with most effective radiation dose on tumor stem cells and tumor cell lines.
Author Response
The authors thank the reviewer for his/her comments. English has been corrected by professional agency (Online English).
The corrections made to the manuscript will be highlighted in yellow in the manuscript.
1.
CuPRiX nanoparticles are issued from AGuIX by releasing gadolinium from DOTAGA chelates surrounding the nanoparticles. The release of the gadolinium is obtained by placing AGuIX nanoparticles in HCl solution and heating at 50°C (concentration of 100 g.L-1 for AGuIX and 1M for HCl). The release of gadolinium increase with the length of the reaction. CuPRiX1 and CuPRiX2 are respectively obtained after respectively 3 and 4 hours of reaction before purification by tangential filtration. The number of uncomplexed chelates around the nanoparticle is finely tuned by varying the length of the reaction.
This has been clarified in the Material and Methods and the Results and Discussion sections.
2.
CuPRiX is the name of the new nanoparticle. Cu is for copper, iX for irradiation X and PR stands for Paul Rocchi first author of this publication. CuPRiX is a general name for nanoparticles issued from AGuIX nanoparticles and displaying DOTAGA chelates after acidic treatment. CuPRiX1 and CuPRiX2 are the nanoparticles obtained after 3 and 4 hours of reaction displaying 182 and 253 nmol.mg-1 of uncomplexed chelates respectively. The definition of CuPRiX1 and CuPRiX2 has been clarified in the text.
3.
The authors want to kindly emphasize that there is a small misunderstanding here. DOTAGA is already present on the AGuIX nanoparticles as stated in lines 250-252. AGuIX nanoparticles are composed of a polysiloxane network on which DOTAGA(Gd) chelates are grafted. Synthesis of AGuIX nanoparticles has already been described in Le Duc et al., Cancer Nanotechnology, 2014. Nevertheless, the authors agree with the referee that it has to be added also in the Material and Methods section. Description has been added in lines 104-106 for more clarity and the reference to the bibliography.
4.
The authors think that there may be a little confusion here. The nanoparticles are ultrasmall and due to the size inferior to 6 nm are eliminated by the kidneys. But it has been shown that AGuIX nanoparticles accumulate in tumors via the EPR effect in more than 10 different animal models (See F. Lux et al. Br. J. Radiol. 2018) but also in humans as recently emphasized in a phase Ib clinical trial on 15 patients suffering from brain metastases (See C. Verry et al., Sci. Adv. 2020 and C. Verry et al., Radiother. Oncol., 2021). EPR effect is mostly described for larger nanoparticles but as recently emphasized by our group in a review, it is also observed for ultrasmall nanoparticles (See G. Bort et al., Theranostics, 2020).
5.
The reasons for choosing lung and head and neck cancer cell lines are outlined in lines 462-466. In addition, the authors have experience in using head and neck cancer stem cells and have previously demonstrated their migration capacity in vitro (Gilormini et al., J. Vis. Exp., 2016; Moncharmont et al., Oncotarget, 2016; Wozny et al., Cancers, 2019). The main scope of the paper was to show the interest of chelating copper while maintaining an efficient radiosensitization (observed on many cell lines for AGuIX nanoparticles including lung and head neck cancer cell lines, See L. Sancey et al., Br. J. Radiol., 2014). Radiosensitization experiment was done using conventional radiation dose for these cell lines.
). Radiosensitization experiments were done using conventional radiation dose for these cell lines (Wozny et al, Br J Cancer, 2017, Detappe et al., Nano Lett., 2017).
Only one fraction (precision added to the manuscript) was used for each irradiation dose and the most effective combination was 4 Gy + nanoparticles for A549 cells and 6 Gy + nanoparticles for SQ20B-CSCs cells).
6.
The authors agree with the referee that radiosensitization of nanoparticles has to be characterized by many different methods but radiosensitization of AGuIX nanoparticles has already been proven in vitro and in vivo on more than 40 cell lines including lund and head and neck cancer cell lines (L. Sancey et al., Br. J. Radiol., 2014; F. Lux et al., Br. J. Radiol., 2018). Clonogenic assay is the gold standard for radiosensitization and is used in this paper to show that CuPRiX and AGuIX seem to have equivalent radiosensitizing effect for same concentration of heavy atoms (i.e. gadolinium). Furthermore, the authors recognize that the formation of ROS within tumor cells is an important parameter in the study of radiosensitization by metallic nanoparticle. Nevertheless, the nanoscale mechanisms underlying metal-based nanoparticles-induced ROS production is poorly understood (Liu et al., Theranostics, 2018). The authors believe that changes in ROS production due to metallic nanoparticles treatment should be studied just after irradiation (during the early chemical phase happening 10-12s after the irradiation) and this is, to their knowledge, not possible to measure biologically. Only simultation performed by physicists could estimate the effect of nanoparticles on early ROS production.
7. The authors agree with the referee that yH2AX would have been an interesting experiment to complete radiosensitization characterization but it has already been shown for AGuIX NPs (L. Sancey et al., Br. J. Radiol., 2014; F. Lux et al., Br. J. Radiol., 2018) and full characterization of the radiosensitization properties of CuPRiX is beyond the scope of this article. Nevertheless, for the same quantity of gadolinium, it should be very similar for AGuIX and CuPRiX as it is observed for clonogenic assay. Further studies with an in-depth characterization of molecular mechanisms will be performed in future experiments but is beyond the scope of this article that is more focused on copper chelation.
Reviewer 2 Report
The paper is devoted to the modification of already known AGuIX nanoparticles and further to the investigation of the biological properties of products obtained. The experimental results provided in the article are certainly of a very high level and verified by relevant techniques including DLS with z-potential measurements. But still, there is one question that should be clarified in the introduction or conclusion section.
Copper is certainly important metal both in the biology as a whole and in the tumor biology in particular. And the results obtained for the copper-mediated LOX enzyme indirectly certify that CuPriX can influence the copper level in biological liquids. But still free DOTA (DOTAGA) group is very strong complexing agent for different metal cations (see. f.e. Coordination Chemistry Reviews Volume 253, Issues 13–14, July 2009, Pages 1906-1925) including Fe, Zn, Ca, Co which is also important bio-metals. Thus it must be clearly stated that influence of CuPriX (compared with AGuIX) on the tumor cells can be explained by DOTA complexation not only with Cu cations.
Author Response
The authors thank the reviewer for his/her kind comments.
The authors agree with the referee that DOTAGA has also strong complexation constants on other divalent cations like Ca2+, Co2+ or Zn2+ but the complexation constant is at least two order of magnitude larger for Cu2+ (log K(Cu(II)) = 22.3; log K(Zn(II)) = 20.8; log K(Co(II)) = 20.2 and log K(Ca(II)) = 17.2) (G. Anderegg et al., Pure Appl. Chem. 2005). Moreover, Zn2+ and Ca2+ need higher chelation quantities to see an important biological effect. The question of iron is more complex as iron (III) is less available in free form in an aqueous medium and higher chelation is required to see an observable effect.
The authors have added in the paper that the chelation of other metals may also play a role.
Reviewer 3 Report
The authors present the development of novel multimodal ultrasmall nanoparticles, called CuPRiX combing copper chelation and radiosensitization. These NPs are based on the clinical AGuIX nanoparticles made of a polysiloxane matrix on which gadolinium chelates are grafted. The authors suggest that these novel hybrid NPs can offer copper chelation and radiosensitization and they evaluated their efficacy in vitro on two cancer cell lines (oral and lung). Moreover, as they referred: “Two patents have been filed on the results described in this publication by PR, FL and OT 513 for the chemical process (FR2011904) and by PR, FL, OT, DBV and CRL for copper chela-514 tion in oncology (FR2011903)”.
It is a very well-written original manuscript and the experimental work was carried out carefully and is well organized. Many techniques such as HPLC–UV, ζ-potential measurements, dynamic light scattering, clonogenic assay etc have been combined to characterize the efficiency of using this type of NPs.
I strongly suggest this manuscript for publication. I have only a comment. I would like the authors to discuss more the cytoxicity of this hybrid NPs with the prospect for in vivo trials.
Author Response
The authors thank the reviewer for his/her kind comments.
The authors agree with the referee that the modification of AGuIX nanoparticles should induce modification of toxicity but it will probably be relatively minor. The authors have already tested the nanoparticles in author cell lines without sign of notable toxicity and multiple injection in tumor bearing animals have not shown evidences of toxicity (data not shown).
Round 2
Reviewer 1 Report
Thanks for addressing all the comments with proper references whereever it was necessary.
best wishes,